# Assessment of the Role of Leptin and Adiponectinas Biomarkers in Pancreatic Neuroendocrine Neoplasms

**DOI:** 10.3390/cancers15133517

**Published:** 2023-07-06

**Authors:** Agnes Bocian-Jastrzębska, Anna Malczewska-Herman, Violetta Rosiek, Beata Kos-Kudła

**Affiliations:** Department of Endocrinology and Neuroendocrine Tumors, Department of Pathophysiology and Endocrinogy, Medical University of Silesia, 40-514 Katowice, Poland; anna.malczewska@sum.edu.pl (A.M.-H.); vrosiek@sum.edu.pl (V.R.); bkoskudla@sum.edu.pl (B.K.-K.)

**Keywords:** pancreatic neuroendocrine neoplasm, leptin, adiponectin, leptin–adiponectin ratio

## Abstract

**Simple Summary:**

Leptin and adiponectin are adipokines involved in the carcinogenesis of gastrointestinal cancers, even though no clear link with pancreatic neuroendocrine neoplasms (PanNENs) has yet been investigated. We aimed to evaluate the possible relationship between leptin, adiponectin, and the leptin–adiponectin ratio, and PanNEN’s grade or stage, including the presence of metastases. We obtained a significantly higher concentration of adiponectin in the control group compared to the study group. The concentration of both adipokines was significantly higher in females than in males. An increased leptin–adiponectin ratio was observed in well-differentiated PanNENs (G1) vs. moderatelydifferentiated PanNENs (G2) and in PanNENs with Ki-67 < 3% vs. Ki-67 ≥ 3%. We identified patients with distal disease as having lower leptin levels and decreased leptin–adiponectin ratio. Thisstudy reveals the importance of adipokines assessment and provides new insights into the mechanisms leading to PanNEN development.

**Abstract:**

Data on the possible connection between circulating adipokines and PanNENs are limited. This novel study aimed to assess the serum levels of leptin and adiponectin and their ratio in patients with PanNENs and to evaluate the possible relationship between them and PanNEN’s grade or stage, including the presence of metastases. The study group consisted of PanNENs (*n* = 83), and healthy controls (*n* = 39). Leptin and adiponectin measurement by an ELISA assay was undertaken in the entire cohort. The serum concentration of adiponectin was significantly higher in the control group compared to the study group (*p* < 0.001). The concentration of leptin and adiponectin was significantly higher in females than in males (*p* < 0.01). Anincreased leptin–adiponectin ratio was observed in well-differentiated PanNENs (G1) vs. moderatelydifferentiated PanNENs (G2) (*p* < 0.05). An increased leptin–adiponectin ratio was found in PanNENs with Ki-67 < 3% vs. Ki-67 ≥ 3% (*p* < 0.05). PanNENs with distal disease presented lower leptin levels (*p <* 0.001) and a decreased leptin–adiponectin ratio (*p* < 0.01) compared with the localized disease group. Leptin, adiponectin, and the leptin–adiponectin ratio may serve as potential diagnostic, prognostic, and predictive biomarkers for PanNENs. Leptin levels and the leptin–adiponectin ratio may play an important role as predictors of malignancy and metastasis in PanNENs.

## 1. Introduction

Pancreatic neuroendocrine neoplasms (PanNENs) are a heterogenous subgroup ofgastroenteropancreatic neuroendocrine neoplasms (GEP-NENs) and account for 30% of GEP-NENs. At least 70% of these tumors are non-functional, well-differentiated, and mostly remain asymptomatic [1]. Symptoms occur when they reach a large size, causing signs of local invasiveness. On the one hand, for that reason, they are often diagnosed at a later stage when the disease is already advanced or metastatic [2]. On the other hand, due to huge progress in diagnostic modalities, the number of PanNENs diagnosed at an early stage is increasing incidentally.

PanNENs, like all NENs, are diagnosed on the basis of histological type, including their differentiation and histological maturity, grading (G), pathomorphological advancement (pTNM), and clinical advancement staging (S). The grade (G) is assessed on the Ki-67 proliferative index and the number of mitotic figures. Ki-67, a cell-proliferation-associated nuclear marker, is used widely in the assessment of the malignant potential of NENs [3]. According to the European Neuroendocrine Tumor Society (ENETS) and the current WHO 2019 classification for GEP-NENs, GEP-NENs are classified in relation toKi-67 into well-differentiated NENs (G1-high mature: Ki-67 < 3%, G2-medium mature: Ki-67 3–20%, G3-low mature: Ki-67 > 20%), poorly differentiated NECs (always G3) and mixed neuroendocrine–non-neuroendocrine neoplasms (MiNENs) [4,5]. The histological and pathological staging of the tumors in patients with PanNENs is based on the assessment of the tumor size (T), the presence of nodal metastases (N), and the presence of distant metastases (M).

The annual prevalence of metastatic neuroendocrine neoplasms (mNENs) is rising. Approximately 40–50% of patients with NENs present distant metastases at the time of diagnosis [6]. Metastases occur in 50–80% of PanNENs, predominantly in the liver [7,8]. PanNENs have a generally better prognosis than pancreatic cancer.Although these tumors tend to be less aggressive than their adenocarcinoma counterpart, they frequently metastasize (14% localized, 22% regional, and 64% distant) and are likely to be diagnosed at an advanced stage [9]. The most common metastasis locations include lymph nodes, liver, and bone. Regional lymph nodes are the most frequent metastatic sites of PanNENs [10]. Hepatic metastases are observed in 1/3 of patients with newly diagnosed PanNENs [11]. Bone metastases occur in 8% of PanNENs [12]. The identification of metastatic disease represents the most important prognostic factor after tumor grading [13].

Leptin and adiponectin, the adipocyte-secreted hormones, mostly produced by adipose tissue cells, play many roles. Besides mediating homeostasis, controlling food intake and energy balance, regulating the immune response, and modulating reproduction, they are also involved in carcinogenesis [14]. Leptin potently induces normal and tumoral cell mitogenesis, growth, and motility [15]. In cancer, leptin was demonstrated to promote mitogenic, antiapoptotic, and proangiogenic pathways which are involved in carcinogenic processes [16,17]. Furthermore, leptin is also positively correlated with increased metastasis [15]. Adiponectin decreases the proliferation of several cell types including cancer cells. Beyond inhibition of tumor cell growth and survival, adiponectin inhibits angiogenesis and supports cell apoptosis [15,17]. Adiponectin may suppress the development of metastasis and low adiponectin concentrations are associated with metastases in the course of multiple cancers [18]. Moreover, adiponectin can antagonize the actions of leptin [17]. Due to the antagonistic effect on carcinogenesis, proper cross-talk between leptin and adiponectin as well as their right proportions, defined as the leptin-to-adiponectin ratio, are crucial [17].

Several studies have shown a link between leptin, adiponectin, and gastrointestinal cancers together with pancreatic cancer [19,20,21]. The associations between leptin and adiponectin levels and PanNENs are still not well documented. Data on the possible connection of circulating adipokines with other gastrointestinal cancers, such as GEP-NENs, are limited. The increasing prevalence of PanNENs [22], the proven connection between leptin, adiponectin, and pancreatic cancer, and the unreported association with PanNENs highlight the importance of achieving a clear understanding of the underlying mechanisms involved in the development of PanNENs. To our knowledge, in the available literature, no similar observations and no studies assessing circulating leptin and adiponectin levels have been undertaken in PanNENs. Based on the evidence, we hypothesize that these adipokines are involved in the pathogenesis and biology of PanNENs. Consequently, in this study, we aimed to evaluate levels of leptin and adiponectin in PanNENs and healthy controls and explore the relationship between the concentrations of leptin, adiponectin, and the leptin–adiponectin ratio in PanNENs and tumor-related characteristics, such as the disease extent (presence of metastases), grade, or Ki-67. The leptin, adiponectin levels, and leptin–adiponectin ratio were also compared in relation to sex.

## 2. Materials and Methods

### 2.1. Study Participants

The study cohort consisted of 83 patients with PanNENs (confirmed pathologically in accordance with the WHO’s 2019 classification and the American Joint Committee on Cancer/Union for International Cancer Control’s (AJCC/UICC) 2017 classification) and 39 healthy controls. All patients were recruited at the Department of Endocrinology and Neuroendocrine Tumors, European Neuroendocrine Tumor Society (ENETS) Centre of Excellence for NENs, Medical University of Silesia, Poland. The healthy controls were family members of the patients attending the department who assessed their health as very good, without any significant morbidity or history of cancers. Informed written consent was obtained from all study participants. Body parameters such as height (cm) and weight (kg) were collected for the entire study cohort. BMI was calculated as weight divided by height in meters squared. Participants were grouped into three BMI categories: normal weight (BMI < 25 kg/m^2^), overweight (BMI 25–29.9 kg/m^2^), or obese (BMI ≥ 30 kg/m^2^). Table 1 provides the demographic and clinicopathological characteristics of the study and control groups.

The inclusion criteria comprised PanNENs being pathologically confirmed (study group) and written informed consent being granted(study and control groups). The exclusion criteria included a lack of informed consent, co-presence or history of other cancers or diseases, such as diabetes, kidney, or hepatic failure, age below 18, pregnancy, or lactation. All study participants underwent physical examination and provided a full history of comorbidities. Patients with TNM stage IV had lymph node, liver, and bone metastases. Surgical treatment included tumor enucleation, pancreatoduodenectomy, distal resection with removal of the spleen, total pancreatectomy, and resection of liver metastases.Most of the patients were treated with somatostatin analogs (octreotide, lanreotide). RLT (PRRT)radioligand therapy (radioisotope therapy) with ^177^Lu or ^90^Y/^117^Lu, molecularly targeted treatment (everolimus, sunitinib), and chemotherapy (CAPTEM: capecitabine + temozolomide, STZ + 5-FU: streptozocin + 5-fluorouracil, FOLFIRINOX: oxaliplatin + irinotecan + 5-FU, PE: cisplatin + etoposide, KE: carboplatin + etoposide) were performed in several patients.

### 2.2. Assessment of Serum Biomarkers

Prior to blood collection, body measurements were taken. The concentrations of adiponectin and leptin were determined in serum obtained from fasting whole blood samples drawn in the morning from the cubital veins. After immediate blood centrifugation, the separated serum was transferredinto Eppendorftubes, anonymously coded, and stored at –80 °C until analysis. Leptin and adiponectin serum concentrations were determined by the immunoenzymatic assay using Human Leptin ELISA (BioVendor, Brno, Czech Republic) and Human Adiponectin ELISA kits (BioVendor, Czech Republic). The determinations were performed according to the manufacturer’s protocol. Results were given as ng/mL (leptin) and μg/mL (adiponectin). The leptin metrics were as follows: the method’s sensitivity was 0.2 ng/mL; intra-assay precision and inter-assay precisions were 4.2–7.6% and 4.4–6.7%, respectively. The adiponectin metrics were as follows: the method’s sensitivity was 0.47 ng/mL; intra-assay precision and inter-assay precisions were 3.3–4.4% and 5.8–6.2%, respectively. The leptin to adiponectin ratio was calculated as leptin in ng/mL divided by adiponectin in μg/mL.

### 2.3. Statistical Analysis

Statistical analysis was performed using Statistica version 13.3, TIBCO Software Inc. (Palo Alto, CA, USA, 2017). Data were tested for normal distribution using a W Shapiro–Wilk test. Results are presented as the means with standard deviations (SD) or medians and interquartile ranges. Groups were compared with a nonparametric Mann–Whitney U test. Correlations between variables were assessed by calculating the correlation coefficient using the R Spearman method. All results and differences were considered as statistically significant with *p <* 0.05 or lower.

## 3. Results

The study and control groups did not differ by BMI.

### 3.1. Leptin, Adiponectin Levels and Leptin–Adiponectin Ratio in the Study vs. Control Group

Adiponectin levels were significantly higher in the control vs. the study group (*p <* 0.001) (Figure 1, Table 2). Leptin levels as well as the leptin–adiponectin ratio did not differ significantly between the study and control groups (Table 2).

### 3.2. Leptin, Adiponectin Levels and Leptin–Adiponectin Ratio in the Study vs. Control Group—Sex Analysis

Leptin levels were significantly higher in females than males in the study vs. control group (*p* < 0.01). Adiponectin levels were significantly higher in females than males in the controls vs. the study group (*p <* 0.01). The leptin–adiponectin ratio did not significantly differ by sex (Table 3).

### 3.3. Leptin, Adiponectin Levels and Leptin–Adiponectin Ratio—Grade Analysis

Leptin and adiponectin levels, as well as the leptin–adiponectin ratio, were assessed by the tumor grade. An increased leptin–adiponectin ratio was observed in grade 1 PanNENs vs. grade 2 PanNENs (Figure 2, Table 4). Adiponectin or leptin levels separately did not differ by grade (Table 4).

### 3.4. Leptin, Adiponectin Levels and Leptin–Adiponectin Ratio—Ki-67 Analysis

Leptin, adiponectin levels, andthe leptin–adiponectin ratio were analyzed by Ki-67%. An increased leptin–adiponectin ratio was observed in PanNENs with Ki-67 < 3% vs. Ki-67 ≥ 3% (Figure 3, Table 5).

### 3.5. Leptin, Adiponectin Levels and Leptin–adiponectin Ratio—Analysis by Disease Extent

Leptin, adiponectin levels, and the leptin–adiponectin ratio were assessed by the disease extent—the study group was categorized into localized, regional, and distal disease groups. PanNENs with distal disease presented lower leptin levels and a decreased leptin–adiponectin ratio vs. localized disease (Table 6, Figure 4).

### 3.6. R Spearman’s Correlation

Serum concentrations of adipokines and the leptin–adiponectin ratio correlated with various variables (Table 7). Adiponectin correlated with BMI (R= −0.36; *p* < 0.001) and chromogranin A (R = 0.36; *p <* 0.001). Leptin correlated with BMI (R= 0.46; *p <* 0.001), chromogranin A (R= −0.23; *p <* 0.05), serotonin (R= −0.20; *p <* 0.05), 5-HIAA (R= −0.20; *p <* 0.05) and Ki-67 (R= −0.27; *p <* 0.001). The leptin–adiponectin ratio correlated with BMI (R= 0.54; *p <* 0.001), chromogranin A (R= −0.34; *p <* 0.001), CEA (R= −0.20; *p <* 0.05) and Ki-67 (R= −0.28; *p <* 0.01).

## 4. Discussion

Obesity-associated abnormalities in the secretion of adipokines such as leptin and adiponectin by adipose tissue can promote cancer development and lead to the activation of oncogenic intracellular molecular pathways [23]. Both adipokines affect obesity-related cancer risk. In a systematic review and meta-analysis by Yeong et al., adiponectin was significantly associated with decreased risk of cancer while leptin was significantly associated with an increased risk of cancer [24]. Only a few papers address the role of obesity in the occurrence of GEP-NENs. Israeli researchers investigated potential relationships between obesity and incidental GEP-NENs. They found that increased height and weight were associated only with the risk of gastric NENs [25]. Three Italian referral centers for NENs proved that independent risk factors for GEP-NENs were a family history of non-neuroendocrine GEP cancer, type 2 diabetes mellitus (T2DM), and obesity. Taking into account the primary site, obesity was confirmed as an independent risk factor for PanNENs and intestinal NENs [26]. According to the results of Santos et al., visceral obesity is also associated with well-differentiated GEP-NENs [27].Barrea et al. suggest that visceral adiposity dysfunction is connected with the worsening of clinicopathological characteristics in GEP-NENs [28].

In a real-world data analysis, Ranallo et al. demonstrated a correlation between the survival outcome and skeletal muscle and adipose indexes. Authors obtained a statistically better progression-free survival (PFS) in mNENs patients on everolimus treatment with increased visceral, subcutaneous and total body fat indices. Their findings may be linked to the mammalian target of the rapamycin (mTOR) pathway involved in adipose tissue metabolism, which inhibits everolimus [29].

The mean BMI in the study group was 24.35 ± 4.14 kg/m^2^ and was similar to the control group (24.19 ± 3.52 kg/m^2^).In the study group, 45 patients presented normal weight. A total of 38 patients were overweight or obese. In the control group, 21 subjectspresented with normal weight. Overweight and obesity were found in 18 subjects. The prevalence of overweight and obese participants was similar in both groups. Leptin, adiponectin, and the leptin–adiponectin ratio were significantly correlated with BMI.The results of studies examining the relationship between BMI and PanNENs are contradictory [30]. There are studies that demonstrate that BMI is higher in patients with PanNENs [31] and insulinoma [32], whereas Hassan et al., as well as Valente et al., observed no significant increased risk among overweight or obese individuals [33,34]. In our study, there was no significant difference between the study and control groups in regard to BMI. On one hand, despite its undeniable advantages, it is notable that BMI is a relatively crude measure of body composition and adiposity and does not differentiate between lean and fat mass [35]. Results of the above-mentioned study [29] highlight the importance of quantifying percentages of muscle and fat, rather than just assessing total weight and BMI. On the other hand, some PanNEN patients might have experienced disease-related weight loss or weight gain, and collecting information about patients’ BMIs before the diagnosis is often impossible [36].

Diabetes increases the risk of cancer through various mechanisms such as hyperinsulinemia and an increase in insulin growth factor 1 (IGF-1) secretion [27,37]. The relationship between glycemic disorders and PanNENs is bidirectional and the mechanisms which link diabetes to the development of PanNENs are not wellknown [38]. Patients with PanNENs have glucose metabolism abnormalities [27], and diabetes significantlyincreases the chance of developing PanNENs [26,30,39]. Halfdanarson et al. found that patients with PanNENs are more likely than control patients to have a history of diabetes and defined diabetes as a risk factor for PanNENs [31]. Additionally, Hassan et al. showed that patients with diabetes had an increased risk of developing PanNENs compared to patients who did not have diabetes [33]. However, a study conducted by Chinese authors identified diabetes as risk factors only for non-functional PanNENs [36]. Furthermore, Valente et al. demonstrated that diabetes was more prevalent in patients with poorly differentiated PanNENs (G3) and with metastatic disease (TNM stage III–IV) at diagnosis [34]. Tan et al. reported that preoperative new-onset diabetes and impaired fasting glucose (IFG) are correlated with aggressive tumor behavior and poor recurrence-free survival (RFS) of patients with non-functional PanNENs [40]. Different conclusions were drawn from study by Kusne et al. In their case–control study, diabetes had no negative impact on the survival of patients with neuroendocrine tumors [41]. Similarly, Pusceddu et al. have not recognized diabetes as a negative prognostic factor [39].

It appears that an indirect mechanism for diabetes involvement in carcinogenesis is its effect on adipokine concentrations. Adverse leptin and adiponectin production reflecting dysfunctional adipose tissue is presented in diabetes. Obese and nonobese patients with T2DM have an imbalanced adipokine profile [42,43]. Leptin and adiponectin regulating glucose metabolism also take part in the development of diabetes. An altered adipokine profile is associated with the risk of T2DM. It is stated that elevated leptin levels are linked to the presence of insulin resistance and reduced adiponectin levels, with increasing levels of fasting blood sugar level promoting the onset of diabetes [44,45,46,47].

In addition, antidiabetes medicines also affect adipokine levels. Metformin inhibits leptin production and stimulates adiponectin production along with the activation of adenosine monophosphate-activated protein kinase (AMPK), which in turn inhibits mTOR [39,48,49]. Sitagliptin, pioglitazone, liraglutide, dapagliflozin, and empagliflozin have been found to reduce leptin levels [46,50]. Pioglitazone and dapagliflozin increase adiponectin concentrations [50,51].

Metformin, with its cancer-preventive effect, improves response to treatment anddecreases the incidence of cancer incidence and the risk of cancer-related mortality [37]. In advanced PanNENs, the use of metformin could have some antitumor effects in treatment and was associated with prolonged PFS [48,52,53].

Pancreatic cancer is associated with obesity and obesity is a risk factor for pancreatic cancer [54,55,56,57]. General abdominal fatness also increases pancreatic cancer risk [58]. A Mendelian randomization study demonstrates a 34% increase in pancreatic cancer risk per SD increase of BMI (4.6 kg/m^2^) [59]. One of the explanations for this dependency is the secretion of pro-inflammatory and pro-cancer cytokines such as leptin [60]. Moreover, excessive amounts of adipose tissue are related to worse prognosis and higher mortality rate in multiple carcinomas, including of the pancreas [61]. In obese patients with pancreatic ductal adenocarcinoma (PDAC),the response to chemotherapy is reduced [62]. On the contrary, complications such as pancreatic dysfunction, advanced progression of disease, weight loss, or cachexia might be underlying factors for decreased leptin levels [17]. Leptin and adiponectin levels are related to the amount of adipose tissue. In case of excess body fat, leptin levels are elevated while adiponectin levels are reduced [15]; thus, in cachexic patients, leptin levels may be reduced and adiponectin levels increased due to weight loss. Indeed, in the case of undernutrition, Bobin-Dubigeon et al.observed a decrease in levels of leptin while adiponectin levels increased [63]. Even relatively modest weight loss has been shown to noticeably reduce leptin blood levels. To increase plasma adiponectin concentrations, weight loss must be greater (>10% weight loss from baseline combined with >10% reduction in visceral fat mass) [64]. Pancreatic cancer patients with cachexia showed a significantly lower leptin level than those without cachexia [21,65,66], while adiponectin levels were conflicting. Adiponectin has been observed to be elevated [21,67], but also no significant difference in adiponectin level has been noticed [65]. However, in our study, BMI in both groups did not differ and patients did not develop cancer-associated cachexia, which is rare in patients with PanNENs, unlike in those with pancreatic cancer [61], which eliminates the impact of such a cause on changes in adipokine concentrations.

Leptin and adiponectin are involved in pancreatic physiology [68,69,70] as well as human pancreatic adipocytes release adipokines [71]; thus, dysfunction of this adipoinsular feedback loop results in metabolic disorders [72]. These adipokines take part in other pathological processes [73,74], including pancreatic cancer formation [75]. Intraductal papillary mucinous neoplasm (IPMN) dysplastic grade, which can progress to pancreatic cancer, correlates with circulating leptin levels [76]. While leptin enhances pancreatic cancer progression through different signaling pathways, adiponectin may exert a protective effect that is rarely observed with other adipokines [77]. The phosphatidylinositol 3-kinase/protein kinase B(PI3K/AKT) signaling pathway is an important mediator in the development and migration of pancreatic cancer associated with obesity and plays a key role in pancreatic cancer cell proliferation and metastasis [78]. Leptin, through activation of the AKT pathway, may contribute to cell proliferation and glucose metabolism of human pancreatic cells [79];the leptin–Notch signaling axis is also involved in pancreatic cancer progression [80]. Mendonsa et al. suggested that leptin, after binding to its receptors present on pancreatic cancer cell lines, contributes to pancreatic tumor growth through activation of the PI3K/AKT pathway [57]. Decreased adiponectin enhances the susceptibility to pancreatic cancer development and is associated with pancreatic cancer growth and dissemination by inhibiting the action of insulin, which can activate PI3K/AKT signaling [78]. According to White et al., increased body weight significantly accelerates the growth of pancreatic cancer in mice and the resulting increase in leptin levels is a potential mechanism linking obesity and pancreatic cancer. Unfortunately, this does not apply to adiponectin, for it the researchers did not observe a difference between its concentration and tumor size or proliferation [81]. Interestingly, calorie restriction is related to delayed PDAC progression and results in increased serum adiponectin and decreased initially deregulated serum leptin in cancer [82].

In our material, the concentration of leptin was similar in the study and control groups.Our findings are in agreement with those obtained by Romanian and Japanese researchers. In a study by Man et al., the serum levels of leptin in PDAC patients were similar to the control patients [83]. Sakamoto et al. found that plasma levels of leptin were not elevated in patients with pancreatic cancer [84]. A Mendelian randomization study conducted by Dimou et al. did not confirm the relationship between leptin levels and the risk of pancreatic cancer [85].

Certain study results of leptin levels in pancreatic cancer are conflicted. Karabulut et al. showed that the baseline serum leptin levels were significantly higher in patients with pancreatic adenocarcinoma than in the control group [86]. A pooled, nested case–control study from three cohorts of middle-aged and older adults identified a significantly increased risk of occurrence of pancreatic cancer in subjects with higher prediagnostic circulating leptin concentrations among those with longer follow-up [87].Another nested case–control study also revealed that higher prediagnostic levels of plasma leptin were associated with an elevated risk of pancreatic cancer, but only among men, not among women [88]. Data from those studies confirm that increased leptin concentrations correlate with pancreatic carcinogenesis. In turn, some authors reported that hipoleptinemia is linked to pancreatic cancer. Through its pro-immunogenic effects, leptin was shown to inhibit the growth of human pancreatic cancer cells [21,89].In a study by Dalamaga et al., lower leptin levels were associated with pancreatic cancer [67]. The pooled data revealed that circulating leptin levels were significantly lower in patients with pancreatic cancer than in those without pancreatic cancer or with precancerous lesions [65]. Similar results obtained by Gąsiorowska et al., together with Colakoglu et al. [90], present patients with pancreatic cancer as compared to controls had significantly lower plasma leptin [90,91], like other findings, which showed that patients with pancreatic cancer had lower circulating leptin levels compared to control patients [89] or to the overall population [92].

The concentration of adiponectin was significantly higher in the control group compared to the study group (*p <* 0.001). Our results are concordant with data from a prospective study on plasma adiponectin and pancreatic cancer risk in five US cohorts where median plasma adiponectin was lower in case subjects vs. control subjects and low prediagnostic levels of circulating adiponectin were associated with an elevated risk of pancreatic cancer [93].We also conclude that hipoadiponectinemia increases pancreatic cancer risk [94]. Our findings show the anti-cancer properties of adiponectin which are widely reported in the literature. Adiponectin can effectively inhibit proliferation and also induce apoptosis in pancreatic cancer cell lines [95,96] indirectly by antagonizing leptin-induced signal transducer and activator of transcription 3 (STAT3) activation [97]. However, published data about adiponectin levels in pancreatic cancer are inconsistent. The correlation between adiponectin and the risk of pancreatic cancer can be negative or positive [98], or adiponectin may have no effect on pancreatic cancer risk [85]. Colakoglu et al. observed no difference between the study group andthe control group in terms of adiponectin levels [90]. In this study, the prospective EPIC adiponectin cohort showed no association with pancreas cancer risk; however, among never-smokers, higher circulating levels of adiponectin were associated with a reduction in pancreatic cancer risk, whereas smokers tended to have non-significantly increased pancreatic cancer risks, with raised adiponectin levels [99]. In contrast, the prospective study, nested within a large cohort, observed an inverse relationship in male smokers between adiponectin levels and the risk of pancreatic cancer [100]. In a pooled nested case-control analysis ofthree cohorts conducted by Nogueira et al.,high-molecular-weight(HMW) adiponectin was inversely associated with PDAC in never-smokers, not associated in former smokers, and positively associated in smokers. Total adiponectin was not associated with PDAC in nonsmokers or current smokers [101]. Therefore, the association of circulating adiponectin levels with pancreatic cancer risk may be influenced by smoking and also depend on the isoform of adiponectin. Genetic variations in the adiponectin pathway may affect pancreatic cancer risk through their effects on circulating adiponectin [102]. Mohamed et al. provided evidence that variants in the adiponectin gene might influence the development and progression of pancreatic cancer [103].

Some reports showed that adiponectin exerts potent anti-apoptotic effects on pancreatic cells and promotes pancreatic cancer [98,104]. In Dalamaga et al.’s study, pancreatic cancer tumor tissue samples showed positive or strongly positive expression of both adiponectin receptors (AdipoRs) [67]. Data indicate that patients with chronic pancreatitis and pancreatic cancer express higher levels of adiponectin compared with healthy individuals [105] or with precancerous lesions [65]. An association between higher adiponectin levels and pancreatic cancer occurs before and after controlling for, among other things, sex and BMI [67]. The opposite results regarding adiponectin were also obtained in a study conducted by Dranka-Bojarowska et al. The concentration of adiponectin was significantly higher in the pancreatic cancer serum samples compared to the chronic pancreatitis (*p <* 0.01) and control (*p <* 0.01) groups. The authors suggest the potential role of adiponectin as a new tumor marker in pancreatic cancer and in differentiating diagnoses between pancreatic cancer and chronic pancreatitis [106].Likewise, in Oldfield et al.’sopinion, adiponectin could be useful in screening for PDAC in individuals newly diagnosed with T2DM [47].

The leptin–adiponectin ratio is a recognized metabolic parameter. It functions as a predictive marker for adipose tissue dysfunction [107,108,109] and as a useful marker for the diagnosis and prevalence of metabolic syndrome [110,111]. Moreover, because leptin and adiponectin have opposing effects in cells, the balance between those two adipokines is essential and the changes in their ratio link obesity and cancer [112,113]. Indeed, it was recently shown that this ratio is positively associated with several cancers [114]. An increased leptin–adiponectin ratio was associated with an increased risk of some types of cancer [108,115] such as breast cancer [113,116], endometrial cancer [114], colorectal cancer [117], pancreatic cancer [118], and prostate cancer [119]. Hence, the leptin–adiponectin ratio can be used as an estimator of risk, a diagnostic marker, and a therapeutic strategy in cancers associated with adiposopathy [109].

In the current study, the leptin–adiponectin ratio was slightly higher in the study group, but without statistical significance; nevertheless, the results are in line with the cited literature, and to the best of our knowledge, this is the first study that revealed a higher leptin–adiponectin ratio in PanNENs in comparison to the control group. The concentration of leptin was significantly higher in females than in males in the study group compared to the control group (*p <* 0.01). The concentration of adiponectin was significantly higher in females than in males in the control group compared to the study group (*p <* 0.01). Our results are consistent with observations that women have higher leptin and adiponectin concentrations than men [120]. Sex differences are probably due to hormone-related differences in the distribution of adipose tissue [121]. Despite the fact that females have a significantly higher leptin–adiponectin ratio than men [121], in this study, the value of the leptin–adiponectin ratio was not significantly affected by sex. Sex affects the incidence, disease prognosis, and mortality of cancers. There are also notable sex differences in therapeutic response [122,123].There is a higher prevalence of PanNENs in men [11], but both the study group and the control group were predominantly female.

Likewise, sex discrepancies on causal effects for pancreatic cancer can be observed with a higher risk in men but not women [59]. Babic et al. concluded that higher prediagnostic levels of plasma leptin were positively associated with an increased risk of pancreatic cancer among men, but not women [88]. Finding sex differences and variants in genetics showed that females might be protected from many cancers, including pancreatic cancer [123].

In our study, both leptin and the leptin–adiponectin ratio correlated with the grade of PanNENs. We measured significantly negative correlations between leptin and Ki-67 (R= −0.27; *p <* 0.001) as well as between the leptin–adiponectin ratio and Ki-67 (R= −0.28; *p* < 0.01). Moreover, the leptin–adiponectin ratio statistically differed between Ki-67 and grading groups. Differentiation of cancer was also analyzed in terms of adiponectin and leptin concentrations in pancreatic cancer. Dranka-Bojarowska et al. [106], in the case of tumors in medium and low degree of differentiation (G2 and G3), observed a significantly higher concentration of adiponectin without a statistically significant correlation with leptin concentration. Our results are consistent with their results in terms of leptin, but in contrast, we did not observe statistically significant correlations and differences in adiponectin concentration depending on Ki-67 and tumor grading, which is also contradictory with the results obtained by Zyromski et al. which reported that serum adiponectin concentration correlated negatively with tumor proliferation [124]. Unfortunately, these studies did not examine the leptin–adiponectin ratio, which was analyzed in prostate cancer.Siemińska et al. and Di Sebastiano et al. reported that in the poorly differentiated cancer subgroup, patients had a higher leptin-to-adiponectin ratio [119,125]. Moreover, men with a higher Gleason score had significantly greater leptin concentrations than those with a lower Gleason score. The obtained results contrast with our observations. We observed higher leptin concentrations (trending towards significance) and statistically significant higherleptin–adiponectin ratios in patients with PanNEN G1 compared to patients with PanNEN G2. Moreover, we observed a statistically significant difference in the leptin–adiponectin ratio—there wasan increased leptin–adiponectin ratio in Ki-67 < 3% versus Ki-67 ≥ 3%. Changes in leptin and the leptin–adiponectin ratio suggested an association with the aggressiveness of the tumor histology.

Correlations with tumor markers were also evaluated. A positive correlation was found between adiponectin concentration and chromogranin A, whereas negative correlations were found between leptin concentration and chromogranin A and serotonin as well as between the leptin–adiponectin ratio and chromogranin A and CEA. These novel results indirectly demonstrate the relationship between the studied adipokines and PanNENs.

Finally, we questioned whether the leptin and adiponectin levels or their ratio showed any differences according to the disease stage. Serum levels of leptin and adiponectin as well as leptin–adiponectin ratio were assessed depending on the presence of metastases. Similar to the results of the analysis with regard to Ki-67% and tumor grading, patients with distal disease presented lower leptin concentration and decreased leptin–adiponectin ratio.

Few studies were in line with our observations [65,67]. In a study ofa group of patients with ovarian cancer, mean leptin concentration was significantly higher in patients with lower stages of advancement (stage I and II compared to stage III and IV) [126]. In contrast with our findings, most publications revealed no correlation between the disease severity and adipokines concentration. In their study, Colakoglu et al. found that there is no correlation between the disease stage and adiponectin and leptin levels [90]. Likewise, White et al. [75] together with Dranka-Bojarowska et al. [106] did not observe a difference between the concentration and proliferation or tumor stage, and Gąsiorowska et al. did not find any significant differences between the serum leptin level and tumor localization and distant metastases [91].

However, others have documented that elevations in leptin were consistent with disease progression. Elevated leptin promotes pancreatic tumor invasion and metastasis [127]. Leptin enhances the invasion of pancreatic cancer through the increase in matrix metalloproteinase13 (MMP-13) production via the Janus*kinase*2(JAK2)/STAT3 signaling pathway. The increased expression of either leptin receptors (ObRs) or MMP-13 was significantly associated with lymph node metastasis and tended to be associated with the TNM stage in patients with pancreatic cancer [89]. In papillary thyroid carcinoma, higher leptin levels were connected with a more advanced clinical stage, but without differences in serum leptin concentrations with respect to tumor size [128].

We failed to demonstrate an involvement of the changes in circulating adiponectin levels in the metastatic process, just as Colakoglu et al. and Warakomski et al., who, in their studies, found that there was no correlation between tumor size or clinical stage of disease and adiponectin [90,128], even though there were reports documenting elevations of adiponectin in metastasis tumors [98,104].

In the current study, we observed differences in the leptin–adiponectin ratio which were in agreement with data from the Spanish review wherean increased leptin–adiponectin ratio correlated with tumor TNM stages in colorectal cancer [117]. While the decrease in adiponectin levels in the study group seems to be beneficial because it is believed to have anti-proliferative and anti-angiogenic properties [126], the decrease in leptin and the leptin–adiponectin ratio in advanced disease may be difficult to interpret. It could be explained by the increase in insulin resistance, which is another risk factor for cancer [21]. In addition, despite the low concentration of leptin in circulation, leptin produced by the adipose tissue surrounding the tumor may provide locally increased levels of pancreatic tumor stimulation, suggesting that the presence of tumor-associated adipose represents an important influence on the tumor microenvironment [17]. We agree with the authors of [119,126] that in order to properly assess the role of leptin and adiponectin in PanNENs, it is necessary to take a closer look at the tumor microenvironment and the crosstalk between them and other hormones and proinflammatory cytokines, which foster carcinogenesis and cancer development and growth, in addition to assessing concentrations.

It seems that it would be beneficial to study the expression of these adipokines receptors in PanNENs tissues. We propose that high receptor expression may be associated with reduced adipokines serum concentrations as a result of the downregulation mechanism. Data on ObRs and AdipoRs expression are different. Some published data describe ObRs [57,79,89,129] and AdipoRs expression [67,95,97] in pancreatic cancer tissues and cell lines, while others report that pancreatic cancer expresses relatively high leptin but not ObRs [130]. Nevertheless, some papers have documented their presence in pancreatic cancers, so further research and their identification may prove crucial for PanNENs.

Serum leptin levels may be a good diagnostic and predictive tool for the response to treatment in pancreatic adenocarcinoma patients. Chemotherapy efficiency could be decreased by leptin [80]. Obesity (characterized by high levels of serum leptin) is functionally incriminated in the pathogenesis of chemoresistance of PDAC [131]. Leptin could impair chemotherapy cytotoxicity and promote chemoresistance in pancreatic cancer [132]. In a study by Karabulut et al., serum leptin levels were significantly higher in gemcitabine-based chemotherapy-unresponsive patients compared with gemcitabine-based chemotherapy-responsive patients. However, serum leptin concentration had no prognostic significance for survival [86].

A leptin–adiponectin ratio assessment in cancer was conducted by Słomian et al. A study with 43 ovarian cancer patients presented no relationship between the disease severity with the response to treatment and the concentration of the adipokines, although the leptin–adiponectin ratio before treatment correlated with a better response to chemotherapy, i.e., the lower the ratio, the better the clinical response, which leads us to consider the leptin–adiponectin ratio as a predictor of clinical response to treatment [126]. Unfortunately, in the current study, response to treatment was not evaluated; therefore, we could not assess the importance of leptin, adiponectin, and the leptin–adiponectin ratio as predictive factors in PanNENs.However, the promising results encourage exploring this aspect in a future study.

Another interesting aspect of investigating the role of leptin and adiponectin in PanNENs is their effects on the tyrosine kinase mTOR pathway, which are as a selective m-TOR pathway inhibitor—everolimus—and as a receptor tyrosine kinase inhibitor (TKI)—sunitinib—boththerapeutic targets in NENs [11]. Patients with increased subcutaneous and visceral adipose tissue may have higher activation of the mTOR pathway [29]. Rapamycin is an inhibitor of the mTOR pathway which inhibits several processes involved in tumor cell proliferation and survival. Rapamycin also significantly reduces pancreatic tumor growth and mTOR-related signaling [133] and can normalize elevated serum leptin levels through the PI3K/mTOR signaling pathway [134]. Consequently, rapamycin-mediated lowering of serum leptin can be used as anti-leptin therapy in obesity-related pancreatic cancer [29,89,135]. Adiponectin has an anticancer effect through the inhibition of mTOR signaling, and thus can be used for cancer treatment, including for PanNENs [136]. Moreover, the adiponentin–adiponectin receptor 1 (AdipoR1) axis may serve as a predictor of tyrosine kinase inhibitor response and could be a potential therapeutic target in the future treatment of metastatic cancers. In a study on renal cell carcinoma conducted by Sun et al., patients with lowAdipoR1 expression in primary tumor tissues were more likely to suffer from progressive disease during tyrosine kinase inhibitor treatment and had decreased PFS and overall survival (OS) compared to those with highAdipoR1 expression [137]. In this manner, potential patients with metastatic PanNENs might also benefit from a combination of adiponectin and sunitinib therapy due to its enhancing effect the therapeutic efficacy; nevertheless, this promising therapeutic target deserves further investigation.

## 5. Limitations of the Study

The present study also has several limitations that need to be acknowledged. First, the study relied on blood samples from a single institution with a single measurement of adipokines. Second, the control group was relatively younger than the study group, and both groups were predominantly female. Third, we did not analyze body composition; thus, we were unable to investigate the associations of adipocytokines with visceral fat vs. subcutaneous fat. Fourth, the impact of the PanNENs therapies used was not considered. Fifth, the main limitation is that our study did not explore the expression of ObRs and AdipoRs in samples and the relationship between them and circulating adipokines levels, which may also be relevant in the pathogenesis and biology of PanNENs. Nevertheless, we recognize that the above limitations need to be addressed in future studies.

## 6. Conclusions

To our knowledge, our study is the first to investigate the relationship between leptin and adiponectin and PanNENs. Our findings provide new insights into the mechanisms leading to PanNENs. Determination of serum leptin and adiponectin levels may be useful in the diagnosis of PanNENs and could serve as a potential prognostic and predictive biomarker for PanNENs, especially by evaluating their levels with other known NEN tumor markers. Leptin levels and the leptin–adiponectin ratio may be important predictors of malignancy and PanNEN spread (metastasis). Changes in the leptin–adiponectin ratio underline the importance of interplay of both adipokines—the imbalance between these adipokines, as well as interactions with the tumor microenvironment and other carcinogenic factors, may play a role in tumorigenesis and aggressiveness programming in PanNENs. Furthermore, adipokines may not be initiators but rather factors that sustain the tumorigenesis process. Nevertheless, further prospective studies are needed to determine whether leptin, adiponectin, and the leptin–adiponectin ratio are causally related to PanNENs.

## Figures and Tables

**Figure 1 cancers-15-03517-f001:**
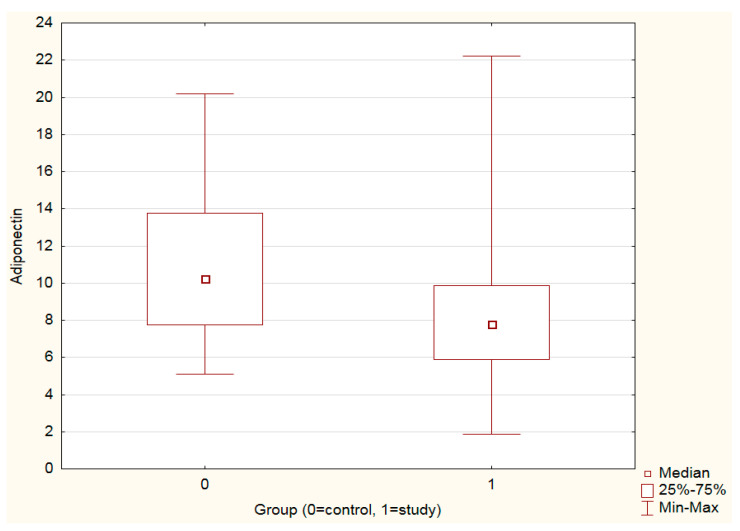
Mann–Whitney U test. Adiponectin levels in the study and control groups: adiponectin levels were significantly higher in controls vs. the study group (*p* < 0.001).

**Figure 2 cancers-15-03517-f002:**
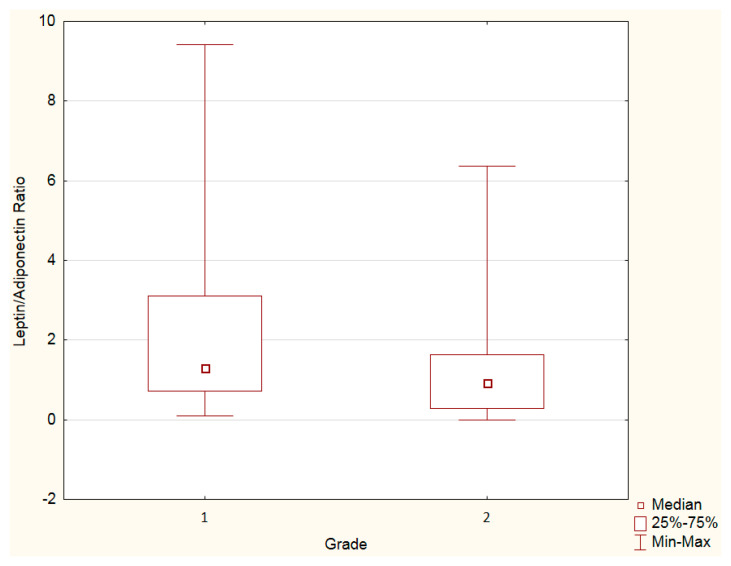
Mann–Whitney U test. Leptin–adiponectin ratio in grade 1 vs. 2 PanNENs: increased leptin–adiponectin ratio was observed in grade 1 PanNENs vs. grade 2 PanNENs (*p <* 0.05).

**Figure 3 cancers-15-03517-f003:**
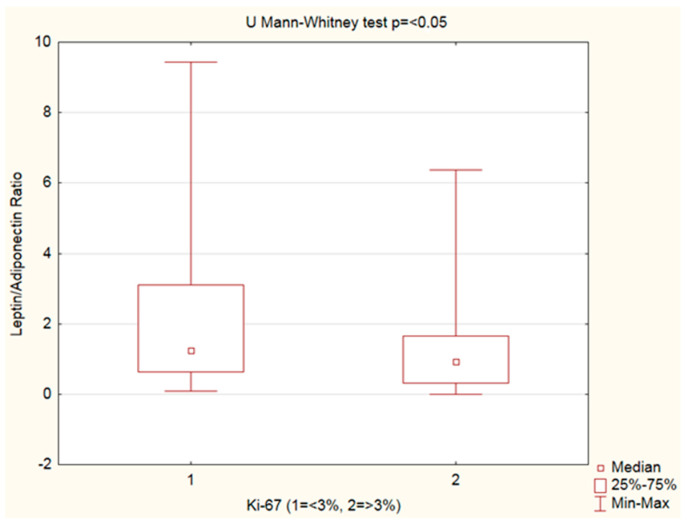
Mann–Whitney U test.Leptin–adiponectin ratio analyzed by Ki-67 groups: increased leptin–adiponectin ratio was observed in PanNENs with Ki-67 < 3% vs. Ki-67 ≥ 3%(*p <* 0.05).

**Figure 4 cancers-15-03517-f004:**
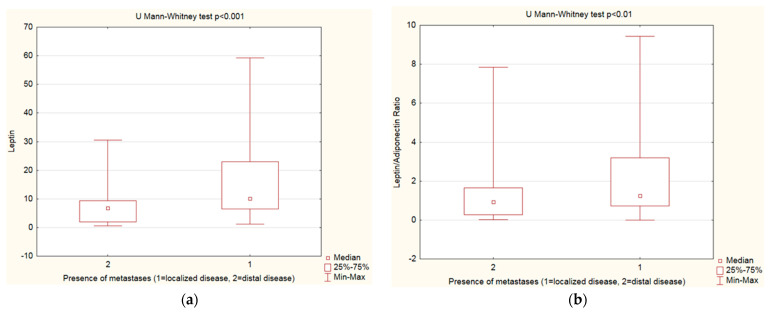
Mann–Whitney U test. (**a**) Leptin levels—analysis by disease extent: PanNENs with distal disease presented lower leptin levels vs. localized disease (*p <* 0.001); (**b**) Leptin–adiponectin ratio analyzed by disease extent: PanNENs with distal disease presented decreased leptin–adiponectin ratio vs. localized disease (*p* < 0.01).

**Table 1 cancers-15-03517-t001:** Demographic and clinicopathological characteristics of study and controls groups.

Variables	Study Group *n* = 83	Control Group *n* = 39
Females/Males	52/31	31/8
Age [years]	50.39 ± 13.79	38 ± 10.52
BMI [kg/m^2^]	24.35 ± 4.14	24.19 ± 3.52
Functionality status		
Non-functional	81	N/A
Functional	2	N/A
Grade		
G1	44	N/A
G2	39	N/A
Ki-67		
<3%	45	N/A
≥3%	38	N/A
TNM stage		
I	27	N/A
IIA	14	N/A
IIB	5	N/A
IIIA	2	N/A
IIIB	8	N/A
IV	27	N/A
Disease extent		
Local	46	N/A
Regional	8	N/A
Distal	29	N/A

mean ± SD, N/A not applicable.

**Table 2 cancers-15-03517-t002:** Leptin, adiponectin levels, and leptin–adiponectin ratio in the study vs. control group.

Variables	Study Group *n* = 83	Control Group *n* = 39	*p* Value ^1^
Leptin (ng/mL)	8.05	7.96	0.822
Adiponectin (µg/mL)	7.79	10.26	<0.001
Leptin–adiponectin ratio	1.03	0.82	0.356

median, distribution of all variables: no-normal; ^1^ Mann–Whitney U test.

**Table 3 cancers-15-03517-t003:** Leptin, adiponectin levels, and leptin–adiponectin ratio in the study vs. control group—sex analysis.

	Study Group			Control Group		
Variables	Females *n* = 52	Males *n* = 31	*p* Value ^1^	Females *n* = 31	Males *n* = 8	*p* Value ^1^
Leptin (ng/mL)	9.63	7.15	<0.01	8.82	7.21	0.414
Adiponectin (µg/mL)	8.37	7.14	0.137	10.97	7.60	<0.01
Leptin–adiponectin ratio	1.20	0.92	0.102	0.72	1.14	0.654

median, distribution of all variables: no-normal; ^1^ Mann–Whitney U test.

**Table 4 cancers-15-03517-t004:** Leptin, adiponectin levels, and leptin–adiponectin ratio—grade analysis.

Variables	PanNENs G1 *n* = 44	PanNENs G2 *n* = 39	*p* Value ^1^
Leptin (ng/mL)	8.88	7.67	0.062
Adiponectin (µg/mL)	7.60	7.89	0.156
Leptin–adiponectin ratio	1.16	0.96	<0.05

median, distribution of all variables: no-normal; ^1^ Mann–Whitney U test.

**Table 5 cancers-15-03517-t005:** Leptin, adiponectin levels, and leptin–adiponectin ratio analyzed by Ki-67 groups.

Variables	Ki-67 < 3% *n* = 45	Ki-67 ≥ 3% *n* = 38	*p* Value ^1^
Leptin (ng/mL)	8.36	7.61	0.108
Adiponectin (µg/mL)	7.21	7.66	0.245
Leptin–adiponectin ratio	1.25	0.93	<0.05

median, distribution of all variables: no-normal; ^1^ Mann–Whitney U test.

**Table 6 cancers-15-03517-t006:** Leptin, adiponectin levels, and leptin–adiponectin ratio analyzed by disease extent.

Variables	Localized Disease *n* = 54	Distal Disease *n* = 29	*p* Value ^1^
Leptin (ng/mL)	10.20	6.81	<0.001
Adiponectin (µg/mL)	7.18	7.89	0.455
Leptin–adiponectin ratio	1.25	0.92	<0.01

median, distribution of all variables: no-normal; ^1^ Mann–Whitney U test.

**Table 7 cancers-15-03517-t007:** R Spearman’s Correlation. Correlations of serum adipokines and leptin–adiponectin ratio with various variables.

Parameter	Leptin (ng/mL)	Adiponectin (µg/mL)	Leptin–Adiponectin Ratio
Leptin (ng/mL)		R = −0.13, *p* = NS	R = 0.87, *p* ≤ 0.001
Adiponectin (µg/mL)	R = −0.13, *p* = NS		R = −0.55, *p* ≤ 0.001
Leptin–adiponectin ratio	R = 0.87, *p* ≤ 0.001	R = −0.55, *p* ≤ 0.001	
BMI (kg/m^2^)	R = 0.46, *p* ≤ 0.001	R = −0.36, *p* ≤ 0.001	R = 0.54, *p* ≤ 0.001
AFP (µg/L)	R = −0.04, *p* = NS	R = −0.06, *p* = NS	R = −0.05, *p* = NS
CA19-9 (U/mL)	R = −0.01, *p* = NS	R = −0.04, *p* = NS	R = 0.01, *p* = NS
CEA (µg/L)	R = −0.15, *p* = NS	R = 0.13, *p* = NS	R = −0.20, *p* ≤ 0.05
CgA (µg/L)	R = −0.23, *p* ≤ 0.05	R = 0.36, *p* ≤ 0.001	R = −0.34, *p* ≤ 0.001
Serotonin (ng/mL)	R = −0.20, *p* ≤ 0.05	R = 0.05, *p* = NS	R = −0.14, *p* = NS
5-HIAA (mg/24 h)	R = −0.20, *p* ≤ 0.05	R = 0.06, *p* = NS	R = −0.19, *p* = NS
Ki-67 (%)	R = −0.27, *p* ≤ 0.01	R = −0.10, *p* = NS	R = −0.28, *p* ≤ 0.01

Abbreviations: BMI, body mass index; SD, standard deviation; AFP, alpha-fetoprotein; CA19-9, carbohydrate antigen 19-9; CEA, carcinoembryonic antigen; CgA, chromogranin A;5-HIAA, 5-hydroxyindoleacetic acid; NS, not significant.

## Data Availability

All data are available upon reasonable request.

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
