# Peer review of "Assessment of the Role of Leptin and Adiponectinas Biomarkers in Pancreatic Neuroendocrine Neoplasms"

_cancers, 2023, doi:10.3390/cancers15133517_

Round 1
Reviewer 1 Report
This manuscript investigates the relationship between circulating leptin/adiponectin and pancreatic neuroendocrine neoplasms (PanNENs). Accordingly, it can potentially benefit future PanNENs patients. Yet, the following additions/clarifications are recommended for the ease of reading and understanding of our future readers.
As indicated in Section 2.1, PanNENs patients having diabetes were excluded from this study. Accordingly, it is suggested to discuss the relationship among leptin/adiponectin, diabetes (especially type II diabetes), and PanNENs in this manuscript.
Also, kindly confirm that the “86” in line 22 is accurate, considering 83 patients were indicated later in Section 2.1 and Table 1. Please further confirm the number “51” in line 243 and “55” in line 244.
In addition, there are two p-value columns in Table 3. I assume one is for study vs control while the other is for male vs female. Kindly confirm this and mark the columns accordingly. Concerning Table 4 and Figure 2, how about healthy controls vs G1 or G2? Further, is any comparison available per TNM stages, such as TNM I vs. TNM IV (optionally vs. healthy controls)?
The following minor changes would be really appreciated.
(1) The result section needs to be renumbered as 3.
(2) The word “sings” in line 42 is believed meant to be “signs.”
(3) A “.” Is missing in line 432.
Reviewer 2 Report
In the present paper, Dr Bocian-Jastrzębska et all assess the serum levels of leptin and adiponectin and their ratio in NEN patients to evaluate the possible relationship between them and grading or prognosis
The topic is suggestive. The paper is quite well written but it needs a moderate english language revision. There are some typos (i.e. page 3 in the table: control Hroups instead Groups)
The exploratory purpose is declared
Some point to address
1. Authors divided patients according Ki67 value but for G2 patients included in the study what is the ki67 range?
2. Why do the authors correlate their findings with CEA AFP and Ca19.9 levels? are they increased in these patients? why? Do the authors have included only NET patients or also MINEN ones?
3.Authors analyze the adiponectin level with the stage. If we look to stage IV patients do you have any data about site of metastases, treatments, therapeutic line , outcome..etc etc? I would like to suggest to add a brief paragraph on this aspect.
4. Do you have any data about the correlation with the outcome?
5. Recently the role of adipose tissue in NEN was investigated ( see the paper : 10.3390/cancers14133231 ) but it is not mentioned in the text. There is a dissertation about the role of different role of visceral and subcutaneous adipose tissue. I think it matchse with your results.
There are some typos. English language needs a moderate revision
Round 2
Reviewer 1 Report
Many thanks for the authors’ amendments and elaboration. Really appreciate them. Yet, I do have one minor concern detailed below.
Further clarification is suggested concerning the p-values in Table 3. The first p-value of the leptin row is marked as <0.01, while the second p-value of the adiponectin row is marked as <0.01. So, my understanding here is that the leptin levels of females vs males in the study group are significantly different, while the adiponectin levels of the females vs males in the control group are significantly different. Kindly confirm whether this understanding is accurate or not. If so, the term “the study vs. control group” in the corresponding descriptions in lines 213-215 is really confusing since I do not see any comparison between study and control here. Similar comments to lines 562-596: does Table 3 compare the control group to the study group? Also, the subject numbers of the control groups in Table 3 need to be corrected.
The following minor changes would be really appreciated.
(1) Is “unrevealed with” in line 104 a grammatical error?
(2) The word “group” in lines 200 and 256 is suggested to be changed to “groups.”
(3) A “.” Is missing in line 393.
(4) It seems an “In” is missing at the beginning of the sentence starting in line 523.
(5) It is suggested to remove either “moreover” or “also” in line 529.
(6) In line 677, it is suggested to either change “expression” to “expressions” or to change “were” to “was.”
Reviewer 2 Report
Authors addressed all the modification required and the paper has been improved